# Family-centered music therapy—Empowering premature infants and their primary caregivers through music: Results of a pilot study

**Barbara M. Menke** [1,2]* *, **Joachim Hass** [2,3], **Carsten Diener**[2], **Johannes Pöschl**[1]*

**1** Department of Neonatology, University Children's Hospital, Heidelberg, Germany, **2** Institute for Applied Research, SRH University Heidelberg, Heidelberg, Germany, **3** Central Institute of Mental Health, University of Heidelberg/Medical Faculty Mannheim, Mannheim, Germany

☙ These authors contributed equally to this work.

* Barbara.Menke@srh.de

**Data Availability Statement:** All relevant data are within the manuscript and its Supporting Information files.

**Funding:** The author(s) received no specific funding for this work.

## Abstract

### Background

In Neonatal Intensive Care Units (NICUs) premature infants are exposed to various acoustic, environmental and emotional stressors which have a negative impact on their development and the mental health of their parents. Family-centred music therapy bears the potential to positively influence these stressors. The few existing studies indicate that interactive live-improvised music therapy interventions both reduce parental stress factors and support preterm infants' development.

### Methods

The present randomized controlled longitudinal study (RCT) with very low and extremely low birth weight infants (born <30+0 weeks of gestation) and their parents analyzed the influence of music therapy on both the physiological development of premature infants and parental stress factors. In addition, possible interrelations between infant development and parental stress were explored. 65 parent-infant-pairs were enrolled in the study. The treatment group received music therapy twice a week from the 21st day of life till discharge from hospital. The control group received treatment as usual.

### Results

Compared to the control group, infants in the treatment group showed a 11.1 days shortening of caffeine therapy, 12.1 days shortening of nasogastric/ orogastric tube feed and 15.5 days shortening of hospitalization, on average. While these differences were not statistically significant, a factor-analytical compound measure of all three therapy durations was. From pre-to-post-intervention, parents showed a significant reduction in stress factors. However, there were no differences between control and treatment group. A regression analysis showed links between parental stress factors and physiological development of the infants.

**Competing interests:** The authors have declared that no competing interests exist.

## Conclusion

This pilot study suggests that a live-improvised interactive music therapy intervention for extremely and very preterm infants and their parents may have a beneficial effect on the therapy duration needed for premature infants before discharge from hospital is possible. The study identified components of the original physiological variables of the infants as appropriate endpoints and suggested a slight change in study design to capture possible effects of music therapy on infants' development as well. Further studies should assess both short-term and long-term effects on premature infants as well as on maternal and paternal health outcomes, to determine whether a family-centered music therapy, actually experienced as an added value to developmental care, should be part of routine care at the NICU.

## Introduction

Premature birth poses an enormous risk not only for the healthy development of premature infants themselves [1–3], but also for the mental health of their parents [4]. Somatic complications during hospitalization [5] as well as stressful sensory overload of the neonatal intensive care unit (NICU) may have negative effects on the early development of the premature infant [6–10]. For example, the acoustic environment in a NICU is not very conducive to the maturation and development of the premature infant [9, 11, 12].

In the absences of a familiar intrauterine sound environment characterized by regularly recurring sounds such as maternal voice and maternal heartbeat [13], the stressful sensory overload in a NICU may result in episodes of apnea, bradycardia and decreased oxygen saturation [14], increased stress hormones [9] as well as sleep deprivation [15] and may also have a negative long-term impact on development [16].

Premature infants have to deal with various, sometimes painful, interventions without the familiar regulating presence of the mother. This early separation can lead to an increased stress level for the infant and the mother [17, 18].

A premature birth is an extreme, tense situation for the parents. Not only due to the objectifiable somatic problems of the infant, but also because of the subjective parental experience [19]. The short gestation period, the premature birth and the resulting early parenting often lead to a feeling of parental insufficiency which impedes parents to adopt the role of primary caregivers [20].

Mothers of preterm infants are at high risk for physiological distress, postpartum depression or anxiety disorders [4, 21]. Mothers who suffer from anxiety and depression are less sensitive to their children's communication and interaction signals [22–24]. The mother-infant interaction and the mother-infant bonding may be impaired during hospitalization, but also 24 months after discharge [24–26]. Symptoms of maternal postpartum depression and anxiety also correlate negatively with the cognitive development of premature infants [27, 28]. Over the last decades, studies focused on stress factors in mothers. However, the few existing studies indicate that fathers as well bear a heavy burden by premature birth [29] and also have to cope with increased stress levels during hospitalization [30, 31]. Fathers of premature infants are confronted with various tasks and roles [32]. They are deeply concerned about the condition of the child, but at the same time also worried about the health of their partner [32]. Fathers often feel responsible for different tasks related to the outpatient context [29, 33].

Family-based interventions and developmental care programs such as Newborn Individualized Developmental Care and Assessment Program (NIDCAP®) [34, 35] or the Entwicklungsförderndes Familienzentriertes Individuelles Betreuungskonzept (EFIB®) (Heidelberg, Germany) [5, 36, 37] draw attention to the importance of accompanying premature infants and their parents at the same time. Moreover, the active involvement of parents as primary caregivers in order to support development, mental health and bonding is a fundamental aspect of both, NIDCAP® and EFIB®.

Along these lines, family-centered music therapy approaches constitute a valuable extension to developmental care programs. As early vocal contact and music have a potential beneficial effect on early development [38], music therapists use interactive live-improvised music therapy interventions to reduce stress, to foster development and to enhance physical and emotional closeness to establish the interaction between parents and children [39, 40]. Previous studies have focused on investigating the effects of music and music therapy interventions on physiological parameters (e.g. stabilization of respiratory rate) in premature infants [41–43] and on the preterm brain development [44]. Until recently, only a few randomized controlled trials have reported data for parents, despite the fact that the integration of parents is described in family-centered music therapy approaches [45–47]. The focus of the above mentioned studies has been limited to reducing maternal anxiety [45, 48–52], the reduction of stress [53], the improvement of the mother-infant interaction [54, 55] and bonding [45, 47, 49]. Until now, studies on whether a reduction in the symptoms of postpartum depression in mothers of premature infants can be achieved using music therapy [56, 57] are rare. Regarding music therapy approaches, only few studies take data from fathers into account [45, 53]. To date, there is no sufficient evidence on the influence of music therapy intervention on stress factors of fathers of premature infants.

Longitudinal studies examining the physiological development of premature infants as well as stress factors in mothers and fathers while taking into account the relationships between these parameters are widely missing in the field of music therapy.

In this pilot study we use a randomized controlled longitudinal study design. The influence of interactive live-improvised music therapy interventions on both the physiological development of premature infants and stress factors in both mothers and fathers as well as possible correlations between infant development and parental stress factors is investigated. We hypothesize that music therapy improves both physiological parameters of the infants and decreases stress factors in mothers and fathers. In particular, we expect infants in the intervention group to exhibit more advanced development at time of discharge as well as shortened length of hospitalization. For the parents, a decrease in ratings of stress, anxiety and depression as well as increased ratings of psychological resources and competence as a primary caregiver are expected during hospitalization of the infant, and in the music therapy group compared to controls.

## Materials and methods

### Study design

The pilot study was conducted in preparation for a randomized controlled trial with a larger sample size comparing a music therapy intervention with treatment as usual as control condition. The Ethics Committee of the Medical Faculty, Heidelberg University approved the study (Study ID: S-044/2016).

The target group of the study was premature infants and their parents or primary caregivers at the beginning and the end of hospitalization of the infant. According to a power analysis (single outcome, two-sample t-test for independent samples), N = 104 parent-infant pairs

would have been needed to obtain a power of 80% assuming medium-sized effects ($\delta = 0.5$). However, this sample size was not feasible given the planned duration of the recruitment process of the pilot study of a maximum of one year. Based on the birth rate of the University Children's Hospital in Heidelberg within the last 12 months, we aimed to initially recruit N = 65 parent-infant pairs and, due to expected drop-out during the course of the study, to include N = 50 pairs in the final data analysis.

Premature infants and their parents were recruited between June 2016 and December 2018 in the Department of Neonatology at the University Children's Hospital, Heidelberg, Germany. The initial recruiting period of one year needed to be extended because of the low rate of admissible parent-infant pairs (see below). Inclusion criteria applied were gestational age at birth ≤ 30 weeks, chronological age ≥10 days of life, clinical stability (at the time of inclusion) as well as sufficient German language skills and informed consent of the parents. Exclusion criteria applied were genetically defined syndrome, neurological diseases (e.g., high-grade intraventricular hemorrhage, periventricular leukomalacia), severe sepsis (necrotizing enterocolitis), clinically relevant malformations as well as malformations that potentially impaired the development of the child up to the age of two years. Preterm infants with need for palliative care were also excluded from the study.

Out of 152 initially recruited parent-infant pairs, 87 infants could not be included in the study: 60 cases did not meet the inclusion criteria and 27 families declined participation. The remaining 65 eligible parent-infant pairs were randomly assigned to the two study groups. In the course of the study, two infants had to be excluded from further analysis because of serious medical complications. One family withdrew from the study because of a misunderstanding regarding the procedure of the study.

At the beginning of the study, data were collected in a pre-post-follow-up design with a fixed number of music therapy interventions, independent of the duration of hospitalization. This initial design was not well accepted by many families who expected interventions throughout the entire hospital stay. Thus, the design was changed during the course of the study to a pure pre-post design (see below for details). The first nine data sets were collected according to the initial design. These data sets were excluded from further analysis, as the two designs differed in schedule, rendering the data sets incomparable to the rest. Only complete data sets were included in data analysis, leading to the exclusion of three data sets containing missing data (one data set in the treatment group and two in the control group).

A total of 50 parent-infant pairs remained as the final cohort to analyze physiological development at discharge. 47 mothers and 30 fathers completed the questionnaires on parental stress factors and were included in the respective analysis.

Using a random sequence, participants were randomly assigned to the treatment or the control group. For this purpose, sealed, opaque, numbered envelopes were used, which were opened only when an envelope was irreversibly assigned to the participant.

## Procedure and intervention

Preterm infants and their families assigned to the control group received standard care as usual (EFIB®) in the NICU of the level III perinatal center at University Children's Hospital, Heidelberg. Premature infants and their families assigned to the treatment group, received an interactive live-improvised music therapy intervention twice a week in addition to the standard care (EFIB®) for time of hospitalization. Fig 1 shows the schedule of the study.

The music therapy intervention was provided by a specialized and certified neonatal music therapist (first author) in accordance with the principles of family-centered care and neonatal music therapy approaches, such as creative music therapy and First Sounds: Rhythm, Breath

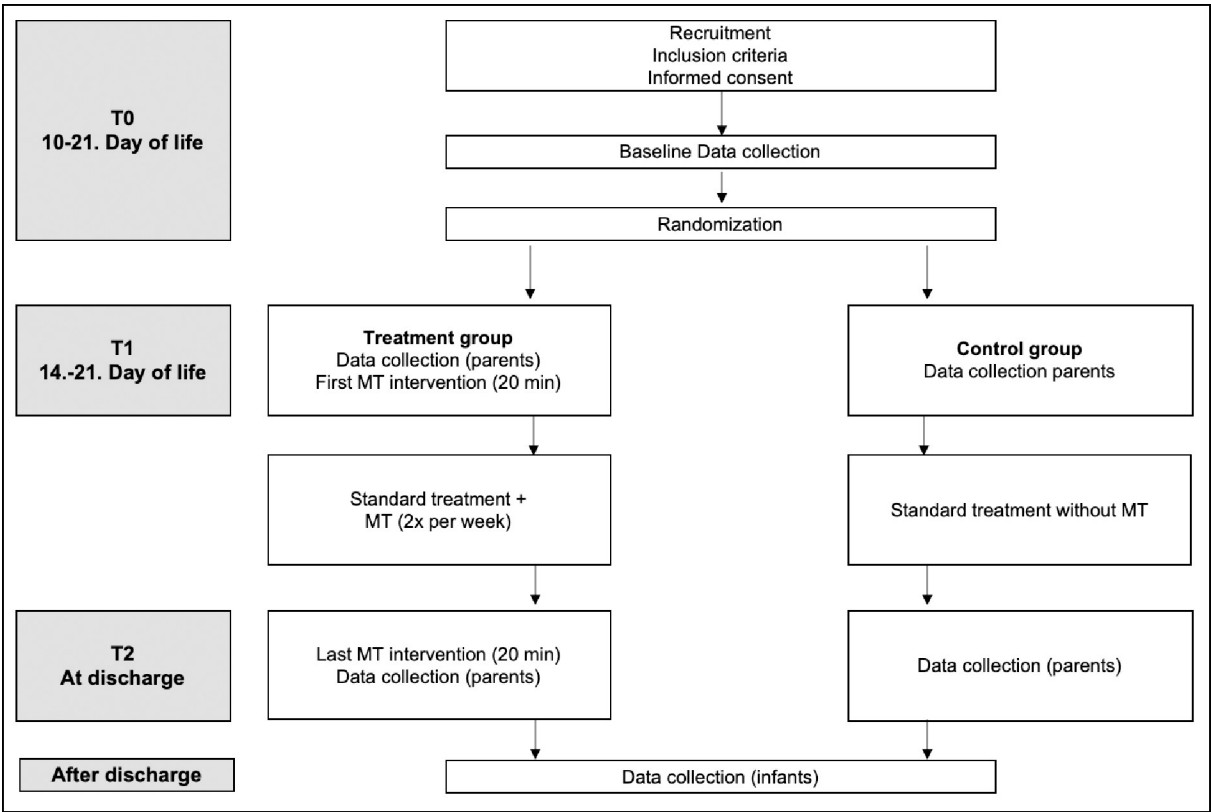

**Fig 1. Schedule of the study.** MT = music therapy.

and lullaby [58–60]. An individual treatment plan was set up based on a parent-infant assessment including parental needs, musical heritage, cultural background as well as the parental willingness to actively participate in the music therapy process. The goals of the music therapy intervention were continuously adapted by the music therapist to the actual individual condition of the infant.

Music therapy sessions were started at the 21$^{st}$ day of life, carried out two times per week for 30 minutes after feeding time. Music therapy interventions were received by infants and parents in skin-to-skin contact or by the infant lying in an incubator. At least six music therapy sessions were received for 20 to 30 min depending on the actual infants' condition and parental needs.

Each music therapy session usually started with an initial touch (at the head and feet of the infant by the music therapists) changing into a therapeutic touch (one hand lays on the chest or the back of the infant). During the period of initial observation, the music therapist gauged increasing or decreasing muscle tension and perceived the infant's breathing rhythm [46]. Then this rhythm was transferred to an infant-directed humming by the music therapist and individually modified aligned to the infant's breathing rhythm, incorporating gestures and facial expressions. To avoid overwhelming by overstimulating the premature infant, the music was kept simple and gentle i.e. continuously flowing but calm tempo, in the character of a lullaby with a high degree of repetitions. Taking the family's musical heritage and culture into account, musical preferences or familiar favorite songs of the parents (song of kin) were considered during the improvisation [59]. To terminate the session, the music therapist faded out the humming and after some more seconds cautiously removed the hands. Right after the

music therapy session, the infant was briefly observed by the music therapist to perceive reactions to the music.

In case parents were involved in the music therapy session, the infant was usually lying on the chest of the parents in skin-to-skin contact. In addition to the music therapy intervention described above, the music therapist often used the monochord to accompany the vocal improvisation [40, 58]. As an instrument specially developed for therapeutic use, a monochord has several strings tuned to the same tone and produces an open, overtone-rich, flowing and relaxing sound as well as intensive vibro-acoustic stimulation. As part of the therapeutic process, parents were invited to relax during the improvisation, encouraged to participate in the vocal improvisation and also empowered to hum or sing to their infant. To foster parent-infant-interaction and bonding, the parents were encouraged to closely observe their infant during the musical intervention, to share their perceptions of infant's signs and reactions. Moreover, parents were also encouraged to use their own voice, e.g., to support their infant in self-regulation and to create closer contact with their child in daily life.

## Data collection and outcome measurement

The primary outcome measure of the study was the physiological development of premature infants at the time of discharge from hospital in terms of body weight, length and head circumference as well as indirect indicators such as the duration of caffeine therapy, the duration of nasogastric/ orogastric tube feed (NGO tube feed) and the length of hospitalization, collected from patient files. As laid out in the Data analysis section, these measures were jointly compared between intervention and control group using Hotelling's $t^2$-test. The resulting multivariate test score provides the key endpoint to assess the effect of family-centered music therapy on the physiological development of the premature infants.

The secondary outcome measure was the reduction of stress, anxiety and postpartum depressive symptoms as well as an increase in parental skills as primary caregivers over the time period from baseline (21st day of infant's life) to postintervention (day of discharge). Changes regarding these measures were recorded for mothers and fathers using self-reporting questionnaires at both stages.

The current level of stress experienced by mothers and fathers was recorded using the *Parental Stress Questionnaire* (PSQ) [61] which assesses stress and resources on a four-point scale i.e. higher scores indicating more stress (possible score of 9–36 points) and more resources (possible score of 6–24 points).

Anxiety was assessed using the German version of the State-Trait-Anxiety-Inventory (STAI) [62]. The questionnaire encompasses 40 items measuring state and trait anxiety on a four-point scale (1–4) with higher scores indicating greater anxiety (possible score of 20–80 points). Postpartum depressive symptoms were assessed using the Edinburgh Postnatal Depression Scale (EPDS) [63, 64]. The self-reported questionnaire comprises 10 items with possible answers on a four-point scale (0–3, possible score of 0–30) with higher scores indicating a higher chance of postpartum depressive symptoms. This questionnaire was only completed by mothers. Parental skills as primary caregiver were assessed using the Parental Competences Questionnaire [65], which comprises eight items for parents to rate their skills on a six-point scale (possible score 8–48) with low scores between 8–24 imply high parental skills.

## Study feasibility

The feasibility of the study was assessed based on the recruitment process and the drop-out rate. Moreover, the number of missing outcome data will serve as the basis for reliable calculation of the sample size for the future full-scale trial.

## Data analysis

Data was analyzed using IBM SPSS Statistics for Windows, version 27 (IBM Corp., Armonk, N.Y., USA). To evaluate the effect of music therapy on the six physiological variables (body weight, length and head circumference at discharge, duration of caffeine therapy, duration of NGO tube feed and the length of hospitalization), we conducted a Hotelling's $t^2$ test, comparing treatment and control group infants. Post-hoc comparisons of each of the six individual variables followed using t-tests for independent samples (with p values adjusted for multiple comparisons by the Bonferroni-Holm procedure) and Cohen's d as effect size measure. Furthermore, we used an exploratory factor analysis (FA) with principal component extraction method to extract the most appropriate factor structure for the six variables. The number of extracted factors was based on eigenvalues ($\lambda \geq 1$, the Kaiser criterion), in agreement with a visual inspection of the scree plot, identifying the factor before the steepest drop in eigenvalues. The resulting two-factor solution was then rotated according to the varimax method, resulting in uncorrelated factors. We compared both factors between treatment and control group using t-tests.

Parental stress factors were assessed with mixed univariate analysis of variance (ANOVA) with the time record (before the start of music therapy and at discharge of the infant) as the within factor and group (treatment vs. control) as the between factor. Finally, a factor analysis was conducted on all parental stress variables (according to the same procedure as described for the factor analysis on the physiological factors above) and the resulting single factor was used as the dependent variable in a linear regression with the physiological factors of the infant as independent variables. Data regarding parental stress factors were analyzed separately for mothers and fathers. In particular, two independent factor analyses were conducted for mothers and fathers. Hypotheses were tested with a significance level of $\alpha = 0.05$. All t-tests were one-sided, reflecting the directed hypotheses, i.e. we expect an improvement of physiological measures and the parental stress factors in the music therapy group rather than merely differences from the control group.

## Results

At the time of birth, the included premature infants (N = 50) had a mean gestational age (GA) of 27 + 3 weeks, (range 23 + 1–30 + 0 GA) and a mean birth weight of 961.2 grams (range 340g – 1,770g). 40.0% (n = 20) of the premature infants were female.

Baseline demographic and clinical characteristics of the study sample are provided in Table 1.

**Table 1.  Baseline demographic and clinical characteristics of the study sample.**

| Variable | MT (n = 24) | CG (n = 26) |
|---|---|---|
| Sex (female) | 12 (50.0%) [a] | 8 (30.8%) [a] |
| Gestational age (days) | 196.29 8.36) [b] | 187.46 (14.80) [b] |
| Birth weight (grams) | 1072.08 (346.47) [b] | 858.85 (334.97) [b] |
| Length (cm) | 36.75 (3.86) [b] | 33.81(4.32) [b] |
| Head circumference (cm) | 25.40 (2.45) [b] | 24.01 (2.72) [b] |
| Mother's age (years) | 33.50 (5.48) [b] | 34.96 (7.51) [b] |

MT = music therapy group; CG = control group

[a] = frequency

[b] = mean and standard deviation.

**Table 2. Differences between groups in physiological development of the infants at time of discharge.**

| Variable (at discharge) | MT (n = 24) | CG (n = 26) | p | Cohen's d |
|---|---|---|---|---|
| Weight (grams)[a] | 2,560.63 (220.09) | 2,602.69 (467.72) | .852[b] | -0.11 |
| Length (cm)[a] | 45.40 (2.01) | 45.00 (2.76) | .852[b] | 0.16 |
| Head circumference (cm)[a] | 32.73 (0.96) | 32.66 (1.51) | .852[b] | 0.05 |
| Duration of caffeine therapy (till dol)[a] | 53.46 (20.87) | 64.58 (24.02) | .205[b] | -0.49 |
| Duration of nasogastric/ orogastric tube feed (till dol) [a] | 63.89 (21.94) | 75.92 (25.53) | .205[b] | -0.51 |
| Duration of hospitalization (days)[a] | 70.79 (21.12) | 86.27 (25.44) | .072[b] | -0.66 |

dol = day of life; MT music therapy group; CG control group

[a] = mean (SD)

[b] = t-test with Bonferroni-Holm correction.

## Physiological development of premature infants

At time of discharge, preterm infants in the treatment group showed descriptively shorter durations of all forms of therapy compared to preterm infants in the control group (Table 2).

We jointly compared all physiological variables between the two groups using Hotelling's $t^2$-test and found no significant difference ($F(6,43) = 1.57$, $p = .181$, $\eta^2_p = 0.18$). Comparing the individual physiological variables across groups also did not reveal any significant differences. Descriptively, infants in the sample receiving music therapy spent 11.1 days less on caffeine therapy, 12.1 days less on nasogastric/ orogastric tube feed and 15.5 days less in hospital, on average (Table 2).

Finally, we exploratively extracted a factor structure using principal component analysis. A two-factor solution emerged, explaining 89% of the total variance: therapy durations (53% variance) and the development of the infant at time of discharge (36% variance). Large scores in the first factor imply longer therapy durations, while large scores in the second factor represent more advanced development. Consistent with the descriptive observations on the individual variables, the scores of the therapy duration factors decreased in the intervention group ($t(48) = -2.06$, $p = .023$, Cohen's d = -0.58), while the scores of the development factor did not differ between the groups ($t(48) = 0.33$, $p = .38$, Cohen's d = 0.09) (Table 3).

## Parental stress factors

**Mothers.** Maternal stress factors (Table 4) were compared between time points T1 and T2 (within factor) as well as between intervention and control group (between factor).

Mixed ANOVAs on each stress factor revealed a decrease of stress levels ($F(1,45) = 20.33$, $p < .001$, $\eta^2_p = 0.34$), state anxiety ($F(1,45) = 11.70$, $p = .001$, $\eta^2_p = 0.21$), trait anxiety ($F(1,45) = 9.63$, $p = .003$, $\eta^2_p = 0.18$) and postpartum depression ($F(1,45) = 29.16$, $p < .001$, $\eta^2_p = 0.39$) at the time of discharge compared to baseline measurement, while resources ($F(1,45) = 5.10$, $p = .029$, $\eta^2_p = 0.11$) and skills as primary caregivers ($F(1,45) = 23.66$, $p < .001$, $\eta^2_p = 0.37$)

**Table 3. Component matrix of a factor analysis of the physiological variables.**

| Factor | Weight | Length | Head circumference | Duration of caffeine therapy | Duration of tube feed | Duration of hospitalization |
|---|---|---|---|---|---|---|
| Factor 1: Therapy duration | 0.26 | -0.03 | 0.03 | **0.96** | **0.98** | **0.97** |
| Factor 2: Development | **0.88** | **0.92** | **0.88** | 0.01 | 0.11 | 0.13 |

Rotated factor loadings for factors extracted by principal component analysis. Loadings above 0.3 are printed in bold.

**Table 4. Descriptive statistics of maternal stress factors.**

| Variable | MT (n = 23) | | CG (n = 24) | |
|---|---|---|---|---|
| | **T1** | **T2** | **T1** | **T2** |
| Stress[a] | 20.87 (3.35) | 18.98 (2.91) | 20.83 (2.97) | 19.32 (1.92) |
| Resources[a] | 19.13 (1.39) | 19.61 (1.47) | 18.37 (2.19) | 19.21 (1.81) |
| State Anxiety[a] | 46.81 (4.40) | 43.02 (7.40) | 46.21 (9.37) | 43.79 (9.17) |
| Trait Anxiety[a] | 42.65 (7.37) | 39.72 (7.16) | 42.04 (11.36) | 38.65 (10.68) |
| Depression[a] | 13.39 (5.11) | 9.61 (4.73) | 10.92 (4.68) | 8.08 (4.71) |
| Skills[a] | 17.60 (3.77) | 14.29 (2.41) | 17.26 (6.33) | 13.95 (4.27) |

MT = music therapy group; CG = control group

[a] = mean (SD).

increased. Contrary to expectations, no difference between intervention and control group and no interactions between the two factors were found (S1 Table).

**Fathers.** As for maternal stress factors, parental stress factors (Table 5) were also compared between time points T1 and T2 (within factor) as well as between intervention and control group (between factor).

As in mothers, the mixed ANOVAs show a decrease of stress levels ($F(1,28) = 25.10$, $p < .001$, $\eta^2_p = 0.47$) and state anxiety ($F(1,28) = 4.89$, $p = .034$, $\eta^2_p = 0.12$) in fathers at the time of discharge compared to baseline measurement, but no changes in trait anxiety ($F(1,28) = 2.85$, $p = .010$, $\eta^2_p = 0.08$), resources ($F(1,28) = 0.02$, $p = .965$, $\eta^2_p < 0.01$) and skills as primary caregivers ($F(1,28) = 0.84$, $p = .368$, $\eta^2_p = 0.03$) were observed between the two points in time. Again, no difference between intervention and control group and no interactions between the two factors were found (S2 Table).

## Relations between physiological development of the infants and parental stress factors

To exploratorily relate the development of the infants with parental stress, we extracted a factor structure of the stress variables using factor analysis, as we did previously for the physiological variables (see Method section for details). We then conducted a regression analysis with the physiological factors as independent and the differences of stress factors between the two points in time as dependent variables.

Both for maternal and parental stress factors, a single-factor solution which explained 40% of variance among the mothers and 44% of variance among the fathers was found. Anxiety

**Table 5. Differences between groups in paternal stress factors.**

| Variable | MT (n = 16) | | CG (n = 14) | |
|---|---|---|---|---|
| | **T1** | **T2** | **T1** | **T2** |
| Stress[a] | 20.69 (3.05) | 18.88 (2.39) | 21.00 (1.96) | 19.14 (1.79) |
| Resources[a] | 19.06 (1.24) | 18.60 (2.27) | 18.67 (1.88) | 19.16 (2.28) |
| State Anxiety[a] | 39.95 (6.76) | 37.82 (9.26) | 40.19 (11.86) | 37.20 (8.14) |
| Trait Anxiety[a] | 34.84 (8.38) | 34.06 (9.48) | 35.21 (10.52) | 33.78 (9.60) |
| Skills[a] | 16.07 (4.54) | 15.40 (4.31) | 14.80 (4.07) | 14.13 (3.04) |

MT = music therapy group; CG = control group

[a] = mean (SD).

**Table 6. Component matrix of a factor analysis of the time differences in the stress variables.**

| Factor | Stress | Resources | State Anxiety | Trait Anxiety | Depression | Skills |
|---|---|---|---|---|---|---|
| Factor 1: Stress differences (mothers) | 0.26 | **-0.60** | **0.81** | **0.77** | **0.74** | **0.39** |
| Factor 1: Stress differences (fathers) | **0.44** | **-0.43** | **0.82** | **0.81** | – | **0.69** |

Factor loadings for factors extracted by principal component analysis (independent for mothers and fathers). Loadings above 0.3 are printed in bold.

and depression were the dominant variables for these factors, while stress itself was less related to the general factors (Table 6). Higher factor scores imply larger ratings on stress factors and smaller ratings on resource ratings.

A multiple regression of the general stress factor with the two physiology factors reveals that the development factor extends a significant negative effect on the stress factor in mothers ($t(42) = -2.5$, $p = .018$), but the therapy duration factor does not ($t(42) = -1.27$, $p = .213$). In fathers, neither of the two factors extends an effect on the stress factor ($t(27) = -0.52$, $p = 0.610$ for development and $t(27) = 0.18$, $p = .862$ for therapy duration).

Finally, we conducted a correlation analysis of the number of music therapy sessions in total and the percentage of sessions with active parent involvement, in the intervention group, with all stress difference variables. No significant correlations between any of these variables were found.

## Feasibility of the study

The duration of recruitment period was 31 months. During this period, 152 premature infants met the inclusion criterion $\leq 30+0$ weeks of gestational age and were examined for further inclusion criteria. The inclusion criteria of the study were not met in 60 cases, mostly because of the instability of the child and the parents' lack of language skills. 27 families declined to participate in the study.

The study was pre-terminated in the following cases: one family withdrew their consent to participate in the study. All data collected up to that point in time were destroyed at the family's request. Two infants from the treatment and control group were discharged before the final observation was feasible. Two mothers from the treatment group and four mothers from the control group did not complete the questionnaire.

All premature infants assigned to the treatment group received at least six music therapy sessions during hospitalization according to the treatment plan (median: 11.17; range: 6–17, total music therapy sessions: 268). No music therapy session had to be prematurely terminated due to overstimulation of the premature infant.

## Discussion

### Feasibility of the study

As one of the first randomized controlled longitudinal (RCT) studies, the aim of this pilot study was to verify the feasibility of the study protocol and to investigate the influence of a live-improvised interactive music therapy intervention for extremely and very preterm infants and their parents on both the infant's development and stress factors in mothers and fathers.

The study protocol was successfully implemented and demonstrates the feasibility of the study. Most of the extremely and very preterm infants were sufficiently stable to participate in the live-improvised interactive music therapy intervention. No reactions of the infant's overstimulation were observed by the certified music therapist. No music therapy session had to be prematurely terminated due to overstimulation of the premature infant. These

results are also confirmed by van Dokkum et al., who made similar observations in their study and consider that an interactive live-improvised music therapy intervention by a certified and specifically trained neonatal music therapist is feasible for extremely and very preterm infants [66].

However, the RCT design entailed various challenges during the implementation process. The recruitment period was significantly longer than anticipated and had to be extended by 19 months (31 instead of originally 12 months planned). 60 of 152 premature infants (39.47%) did not meet the inclusion criteria for the study. Parents refused to participate in the study since they were afraid of being assigned to the control group not receiving music therapy interventions. Still, the moderate rejection rate of 17.76% indicates a high acceptance level and suggests that parents have high expectations about the music therapy intervention with regard to the development of their infants [67]. As a consequence, the compliance of the parents was higher in the treatment group. These findings need to be taken into account for the planning of future RCT at larger scale, especially with regard to the calculation of the sample size. In addition, incentives should be considered in order to motivate participation despite the probability of being assigned to the control group.

## Interpretation of study results

The hypotheses of the present pilot study stated that a live-improvised interactive music therapy intervention supports and promotes premature infants in their physiological development, reduces length of hospitalization as well as parental symptoms of stress, anxiety and postpartum depression, and improves parental skills as a primary caregiver. The hypotheses were partially confirmed. In the following, the results are discussed in detail.

**Physiological development of the infants.** While group comparisons showed no significant difference in the two compound measures of physiological development, group comparisons of the individual variables revealed a significant reduction in the duration of caffeine therapy, the duration of nasogastric/ orogastric tube feed, and the length of hospitalization in the group of infants receiving music therapy.

By entrainment of breathing pattern and vocal improvisation, music therapy supports the regulation and the stabilization of premature infant's respiratory rate [53, 68]. Since a regular respiratory rate favors a stable heart rate, the risk of apnea episodes may be reduced. This in turn may result in a shorter duration of caffeine therapy. These findings confirm results of a meta-analysis that demonstrated an effect of music therapy on the respiratory rate of premature infants (mean difference: -3.91 / min; 95% CI [-7.8 - -0.03] [41]. Since the present study is the first to investigate the duration of caffeine therapy as an outcome in the context of music therapy interventions, results from previous studies are not available for comparison. These first results are promising and should be examined in more detail as part of future studies with a larger sample size.

Music therapy may facilitate stable and regular breathing, which might support preterm infants to develop and to sustain a suck-swallow-breath rhythm. Therefore, music therapy might have an indirect influence on the duration of the NGO tube feed through the direct regulating effect on the respiratory rate.

Successful completion of the caffeine therapy and the capability of taking in self-sufficient calories for growth are important clinical factors for discharge from hospital. Thus, music therapy might have an impact on the duration of hospitalization. In the light of continuously rising health care costs, a shortened length of hospitalization constitutes a valuable benefit from an economic perspective. Further research on the economic impact of music therapy interventions in neonatology hospitals is needed.

The fact that the variables directly related to the infant's development (weight, length and head circumference at the time of discharge) did not differ between groups may be a consequence of the research design: The decision to discharge an infant from hospital is made once its condition stabilizes, which requires a certain degree of physiological development. In this way, a considerable amount of variance is taken out of the developmental variables by choosing the time of discharge as the measurement time. Using this design, hospitalization duration may serve as the most direct proxy for physiological development of the infant. The fact that music therapy has the strongest influence in this variable supports the view of a beneficial effect of this form of therapy on the physiological development of premature infants.

**Parental stress factors.** At time of discharge from hospital, mothers of the treatment group showed a statistically significant reduction in stress, anxiety and postpartum depression. At the same time, they showed increase in their maternal competencies. Fathers of the treatment group also showed a statistically significant reduction in stress and state anxiety. However, music therapy showed neither main nor interaction effects.

These results are in contradiction with findings from previous studies, e.g. parents showed significantly fewer stress signals [53], a reduction in maternal state anxiety [41, 45] and lower symptoms of postpartum depression [56, 57] after music therapy intervention.

It is very likely that an improvement in the medical condition of the infants has the strongest effect on reducing stress and anxiety of parents. Stabilizing the infant's condition by supporting its self-regulation, e.g. by music therapy interventions, might also decrease stress and anxiety levels of the parents. This hypothesis is supported by a regression analysis of the factor scores resulting from an exploratory factor analysis of both the physiological variables of the infant and the variables related to parental stress. In the former, a two-factor solution differentiated between the variables related to therapy duration and those of direct physiological development. In the latter, a single general factor was found encompassing all variables of maternal and paternal stress. The regression revealed a significant negative influence of the development factor on both maternal and paternal stress factors, while no such effect could be found for therapy duration. This result may also explain the missing effect of music therapy on parental stress factors, as infants receiving music therapy needed shorter therapy durations but did not differ from infants in the control group regarding physiological development. So again, the fact that the measurements are made at the time of discharge instead of a predefined point in time may have masked potential benefits of music therapy on parental stress, as it may be mediated by the infants' development.

The results of a qualitative study suggest that music therapy interventions may also directly empower the parents by reducing their stress levels, promoting relaxation and enhancing their well-being [46, 69]. After preterm birth, parental self-confidence and self-efficacy are often covered up by high levels of stress and anxiety. Active participation in music therapy contributes to the empowerment of the parents by the experience of close interaction with their infant. It also delivers support to uncover their intuitive parental capacities. As a consequence, music therapy may have the potential to strengthen parental skills, thereby increasing the quality of parent-child interaction [60] and improving the neurological outcome of premature infants [70]. However, in contrast to these considerations, our study did not reveal any effect of the number music therapy sessions involving parent participation.

The primary attention in this study was on experiences of mothers of premature infants while working with parents in the NICU. However, the results of the present study at baseline underpins that fathers experience the same level of stress as mothers of premature infants. Interestingly, the anxiety levels reported by fathers are lower compared to these reported by mothers. One likely explanation is related to traditional role expectations such as tight control of emotions [71] or different coping strategies of mothers and fathers [72]. The selected music therapy approach needs to be flexible adapted to the different needs of mothers and fathers in

order to provide an adequate family-centered music therapy service. At the University Children's Hospital in Heidelberg, a music therapy service inviting fathers to sing lullabies for their infants offered at evening time was highly appreciated.

Despite various family-centered care measures, parents still showed significant stress levels at time of discharge from hospital. 53.2% of mothers reported clinically significant levels of state anxiety and 29.8% of mothers had clinically significant symptoms of postpartum depression. This might be explained by the perspective of starting to take responsibility for the child on their own. These findings underline the need for family-centered early intervention programs following the NICU discharge. The development and pilot implementation of outpatient music therapy service to continuously sustain a supportive environment with respect to auditory exposures and experiences should be examined in follow-up studies [73, 74].

## Limitations

The study shows several limitations. First: One challenge is related to the complexity of family-centered interventions, encompassing many different procedures making it difficult to draw explicit conclusions. Still, the findings of the presents study suggest benefits for premature infants from a music therapy intervention. For future investigations, an improvement of statistical power is recommended as well as adding further measurements at predefined points in time. Using such fixed-time measurements, it should be possible to disentangle possible effects of music therapy on both physiological development of the infant and the therapy durations needed to achieve these developments. As the reduction of parental stress factors may be influenced by the infants' development rather than therapy duration, such a change in research design might also reveal positive effects of music therapy on these stress factors.

Second: Although both parents were invited to participate in the music therapy session, the participation of both was not feasible in all cases due to clinical practice. This limits the significance of the conclusions about the influence of the music therapy intervention on the reduction of stress, especially for fathers.

Third: The music therapy intervention was offered over the entire inpatient period. Even being beneficial for establishing a therapeutic relationship, this process-oriented approach is very vulnerable to different influencing in- and outpatient factors such as condition of the infant, varying parental engagement as well as demanding expectations of other family members, friends and colleagues.

## Conclusion

This pilot study suggests that a live-improvised interactive music therapy intervention for extremely and very preterm infants and their parents has a beneficial effect on the therapy duration needed for premature infants before discharge from hospital is possible. The results show that the study protocol could be implemented successfully, which demonstrates the feasibility of the study. The pilot study provides valuable information regarding the recruitment process as well as compliance of parents, which can be used for sample size calculation and planning of a future RCT study. Further studies should assess both short-term and long-term effects on premature infants as well as on maternal and paternal health outcomes, to determine whether a family-centered music therapy actually experienced as an added value to developmental care should be part of routine care at the NICU.

## Supporting information

**S1 Table.** a. Main effects of within factor time on maternal stress factors. b. Main effects of between factor group on maternal stress factors. c. Interaction effects of within factor time and

between factor group on maternal stress factors.
(DOCX)

**S2 Table.** a. Main effects of within factor time on paternal stress factors. b. Main effects of between factor group on paternal stress factors. c. Interaction effects of within factor time and between factor group on paternal stress factors.
(DOCX)

**S3 Table.** a. Minimal dataset physiological development infants. b. Minimal dataset maternal stress factors. c. Minimal dataset paternal stress factors.
(PDF)

## Acknowledgments

The authors wish to acknowledge the University Children's Hospital, Department of Neonatology, Heidelberg for their clinical support and thank all families who participated in the study. Acknowledgments are also due to Sven Garbade and Milena Borchers for their valuable comments on the statistical analysis.

## Author Contributions

**Conceptualization:** Barbara M. Menke, Johannes Pöschl.

**Formal analysis:** Barbara M. Menke.

**Investigation:** Barbara M. Menke.

**Methodology:** Barbara M. Menke, Joachim Hass, Carsten Diener.

**Supervision:** Johannes Pöschl.

**Writing – original draft:** Barbara M. Menke, Joachim Hass.

**Writing – review & editing:** Carsten Diener, Johannes Pöschl.

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
