## [Decision Letter · Decision Letter 0]

20 Oct 2020

PONE-D-20-11446

Family-centered music therapy - empowering premature infants and their primary caregivers trough music: A randomized, controlled pilot trail

PLOS ONE

Dear Dr. Menke,

Thank you for submitting your manuscript to PLOS ONE. After careful consideration, we feel that it has merit but does not fully meet PLOS ONE’s publication criteria as it currently stands. Therefore, we invite you to submit a revised version of the manuscript that addresses the points raised during the review process.

The reviewer pointed out several issues in methodology. These concerns need to be addressed carefully as methodology is of key importance in the acceptance criteria in the journal.

We look forward to receiving your revised manuscript.

Kind regards,

Olivier Baud, MD, PhD

Academic Editor

PLOS ONE

Journal Requirements:

Reviewers' comments:

Reviewer's Responses to Questions

**Comments to the Author**

1. Is the manuscript technically sound, and do the data support the conclusions?

Reviewer #1: Partly

2. Has the statistical analysis been performed appropriately and rigorously? 

Reviewer #1: Yes

3. Have the authors made all data underlying the findings in their manuscript fully available?

Reviewer #1: Yes

4. Is the manuscript presented in an intelligible fashion and written in standard English?

Reviewer #1: No

5. Review Comments to the Author

Reviewer #1: The study is original and aims to assess the combined effect of music therapy on premature infants, their mothers, and fathers. Previous studies have focused on assessing the effect of music therapy on premature infants and their mother only. Few studies assessed the impact on the fathers. Here, the paper is presenting the results of a pilot study for the implementation of a future bigger randomized and controlled trial.

General remark: the language needs to be verified carefully, there are some English spelling errors in the manuscript (title, line 62, line 526, line 544…).

1. Abstract:

In the results, the presentation is misleading as you wrote that durations of caffeine therapy, hospital stay, and NGO tube feed were shortened. In fact, reductions were not statistically significant, so please provide 95% confidence intervals around the reduction in days and/or p-values to moderate the findings. The reduction of stress factors in the parents from the treatment group only is also problematic as it does not say anything on the effect of intervention. You need to refer to the control group as well to compare the reduction to no intervention. Results presented in the abstract are generally very imprecise and provide false signals. The study does not provide some evidence that family-centered music therapy had an effect on the premature infant’s development, or on the parental stress factors.

2. Introduction:

Please revise your references list. It seems you have missed some recent and important works published by the team at the University hospitals of Geneva.

3. Methods:

• The investigators are not clear enough about what is primary versus secondary outcomes. They need to choose one key endpoint to be used to design the global study and to test their research hypothesis about a clinical important effect of a family-centered music therapy intervention in premature infants hospitalized at NICUs and their parents. The study will not be sufficiently powered to answer multiple research questions.

• The paragraph on the sample size estimation is not informative enough and it would need to be rewritten. How did you estimate that 50 parent-infant pairs would be necessary? There is no hypotheses presented here.

• Correlation coefficients should be interpreted using specific also subjective scales (for e.g. use Hinkle DE, Wiersma W, Jurs SG. Applied Statistics for the Behavioral Sciences. 5th ed. Boston: Houghton Mifflin; 2003), and not on p-values that are testing the null hypothesis rho=0.

• Normally, intention-to-treat analysis is performed in superiority trials, and completed by per-protocol analysis.

4. Results:

The general presentation of results is not informative enough and needs to be improved. The study shows no evidence for an impact of the intervention on the premature infant’s development. Moreover, the effect of the intervention was assessed by indirect indicators, such as the duration of caffeine therapy, the naso/oro-gastric tube feed, and the duration of hospital stay. The study results are inappropriately reported: there was no statistical difference between both study arms regarding the various primary outcomes assessed, nor regarding the parental factors (all p-values>0.05).

In the text, some useless results are reported and could be deleted, such as the values of test statistics, or the number of degrees of freedom for each statistical test performed.

The absence of evidence for differences in the comparisons of secondary outcomes between both arms does not mean that they are similar. This interpretation should be deleted.

All results on correlations are not correctly presented. The authors have copied the tables provided directly by the statistical software: correlation coefficients and p-values that are not informative here. Moreover, as usual the tables of correlations are symmetric, meaning that results are presented twice in the tables. Another way of presentation should be appropriately provided in a new version (figure? Informative table?). Finally, the interpretation of correlation coefficients is usually qualitative (various scales exist, e.g. in medicine )

5. Discussion:

Globally, the discussion could be shortened and more concise. No new results are expected in discussion (e.g. lines 480-487). Under limitations, the authors have mentioned some modification in the study design but it seems, this was not explained before? Based on the lack of evidence for an effect of family-centered music therapy provided by the study, the authors need to moderate their conclusion.

6. PLOS authors have the option to publish the peer review history of their article (what does this mean?). If published, this will include your full peer review and any attached files.

Reviewer #1: No

---

## [Author Response · Author response to Decision Letter 0]

17 Dec 2020

We would like to thank the reviewers for their careful reading and appreciation of our manuscript, and their helpful comments and critiques. Please find our detailed point-by-point reply to the comments and a list of all changes made to meet the reviewers’ requests below. 

1) Language editing

 “the language needs to be verified carefully, there are some English spelling errors in the manuscript (title, line 62, line 526, line 544…)”

We have carefully reviewed and edited the manuscript’s language and hope that the corrected version holds up to the reviewer’s standards. 

2) Significance of results

“In the results, the presentation is misleading as you wrote that durations of caffeine therapy, hospital stay, and NGO tube feed were shortened. In fact, reductions were not statistically significant, so please provide 95% confidence intervals around the reduction in days and/or p-values to moderate the findings. 

[…]

Results presented in the abstract are generally very imprecise and provide false signals. The study does not provide some evidence that family-centered music therapy had an effect on the premature infant’s development, or on the parental stress factors.

[…]

The general presentation of results is not informative enough and needs to be improved. The study shows no evidence for an impact of the intervention on the premature infant’s development. Moreover, the effect of the intervention was assessed by indirect indicators, such as the duration of caffeine therapy, the naso/oro-gastric tube feed, and the duration of hospital stay. The study results are inappropriately reported: there was no statistical difference between both study arms regarding the various primary outcomes assessed, nor regarding the parental factors (all p-values>0.05).

[…]

Based on the lack of evidence for an effect of family-centered music therapy provided by the study, the authors need to moderate their conclusion.”

Indeed, the data analysis presented in the original version did not contain significant results and we agree that we should not have presented the descriptive differences in a way that suggested significance. For the preparation of the revised version, we sought assistance from a statistician (Joachim Hass, now listed as co-author) and found that we made a mistake in selecting cases to include in the analyses: As mentioned in the manuscript (line 236), “only complete data sets were included in data analysis”, but inconsistently with this statement, we excluded missing data for each comparison individually. Removing all data sets with any physiological data missing led to an exclusion of three more data sets (one dataset in the treatment group and two in the control group). A re-analysis of this more rigidly cleaned data set revealed that the duration of caffeine therapy, NGO tube feed and hospitalization were significantly reduced in the music therapy group. We emphasize that we did not change the exclusion protocol post-hoc with the goal of achieving significance, but only improved adherence to the predefined protocol. In the light of the statistically significant results, we are able to uphold the conclusions we have previously jumped to.

3) Presentation of parental stress factors

“The reduction of stress factors in the parents from the treatment group only is also problematic as it does not say anything on the effect of intervention. You need to refer to the control group as well to compare the reduction to no intervention.”

We agree that we did not adequately differentiate between control and treatment group in the presentation of parental stress factors. We now employed a mixed analysis of variance with time as the within factor and group as the between factor to strictly disentangle these two factors. 

4) References

“Please revise your references list. It seems you have missed some recent and important works published by the team at the University hospitals of Geneva.”

Indeed, we missed some recent works published by the team at the University hospitals of Geneva. We added the following publications to the references list:

Filippa M, Lordier L, De Almeida JS, Monaci MG, Adam-Darque A, Grandjean D, et al. Early vocal contact and music in the NICU: new insights into preventive interventions. Pediatr Res 2020;87(2):249–64. 

Saliba S, Esseily R, Filippa M, Kuhn P, Gratier M. Exposure to human voices has beneficial effects on preterm infants in the neonatal intensive care unit. Acta Paediatr 2018;107(7):1122–30. 

Lordier L, Meskaldji D-E, Grouiller F, Pittet MP, Vollenweider A, Vasung L, et al. Music in premature infants enhances high-level cognitive brain networks. Proc Natl Acad Sci 2019;201817536.

5) End point of the study

“The investigators are not clear enough about what is primary versus secondary outcomes. They need to choose one key endpoint to be used to design the global study and to test their research hypothesis about a clinical important effect of a family-centered music therapy intervention in premature infants hospitalized at NICUs and their parents. The study will not be sufficiently powered to answer multiple research questions.”

Initially, we deliberately chose a number of possible endpoints in order to identify the best possible endpoint for the global study. However, we agree that we need to formulate a single main hypothesis as a goal for this study. We chose to combine the six physiological measures using multivariate mean comparison (Hotelling t2 test, in particular) and principal component analysis. Comparison of these two compound measures between groups is complemented by t tests on the six individual variables and an exploratory factor analysis.

6) Sample selection

“The paragraph on the sample size estimation is not informative enough and it would need to be rewritten. How did you estimate that 50 parent-infant pairs would be necessary? There is no hypotheses presented here.”

[…]

Under limitations, the authors have mentioned some modification in the study design but it seems, this was not explained before?”

The paragraph on the sample size estimation was completely revised and hypotheses were added. The modification in the study design is now explained in detail. 

7) Correlations

“Correlation coefficients should be interpreted using specific also subjective scales (for e.g. use Hinkle DE, Wiersma W, Jurs SG. Applied Statistics for the Behavioral Sciences. 5th ed. Boston: Houghton Mifflin; 2003), and not on p-values that are testing the null hypothesis rho=0.

[…]

All results on correlations are not correctly presented. The authors have copied the tables provided directly by the statistical software: correlation coefficients and p-values that are not informative here. Moreover, as usual the tables of correlations are symmetric, meaning that results are presented twice in the tables. Another way of presentation should be appropriately provided in a new version (figure? Informative table?). Finally, the interpretation of correlation coefficients is usually qualitative (various scales exist, e.g. in medicine )”

To make the manuscript more concise and focused, we decided to remove the full analysis of the correlations and replace it with a regression analysis using the factor structure of the physiological data as well as the parental stress factors. 

8) Intention-to-treat analysis

“Normally, intention-to-treat analysis is performed in superiority trials, and completed by per-protocol analysis.”

Indeed, normally, intention-to treat analysis is common in RCTs. Due to the strong exploratory character, the authors decided against an intention-to-trat analysis. Only complete data sets were included in data analysis. 

9) Presentation of results

“In the text, some useless results are reported and could be deleted, such as the values of test statistics, or the number of degrees of freedom for each statistical test performed.

[…]

The absence of evidence for differences in the comparisons of secondary outcomes between both arms does not mean that they are similar. This interpretation should be deleted.”

We have edited the manuscript to follow these guidelines. In particular, only tests with significant results are presented in the text and test statistics and degrees of freedom are only reported if they do not appear anywhere else in the text. The full details of the analysis have been moved to supplementary tables.

10) Discussion

“Globally, the discussion could be shortened and more concise. No new results are expected in discussion (e.g. lines 480-487).“

The discussion was revised and has been shortened and formulated more precisely. No new results are mentioned in the discussion, yet.

We look forward to receiving your response.

Kind regards,

Barbara Menke

---

## [Decision Letter · Decision Letter 1]

28 Jan 2021

PONE-D-20-11446R1

Family-centered music therapy - empowering premature infants and their primary caregivers through music: Results of a pilot study.

PLOS ONE

Dear Dr. Menke,

Thank you for submitting your manuscript to PLOS ONE. After careful consideration, we feel that it has merit but does not fully meet PLOS ONE’s publication criteria as it currently stands. Therefore, we invite you to submit a revised version of the manuscript that addresses the points raised during the review process.

We look forward to receiving your revised manuscript.

Kind regards,

Olivier Baud, MD, PhD

Academic Editor

PLOS ONE

Reviewers' comments:

Reviewer's Responses to Questions

**Comments to the Author**

1. If the authors have adequately addressed your comments raised in a previous round of review and you feel that this manuscript is now acceptable for publication, you may indicate that here to bypass the “Comments to the Author” section, enter your conflict of interest statement in the “Confidential to Editor” section, and submit your "Accept" recommendation.

Reviewer #1: (No Response)

2. Is the manuscript technically sound, and do the data support the conclusions?

Reviewer #1: Partly

3. Has the statistical analysis been performed appropriately and rigorously? 

Reviewer #1: I Don't Know

4. Have the authors made all data underlying the findings in their manuscript fully available?

Reviewer #1: Yes

5. Is the manuscript presented in an intelligible fashion and written in standard English?

Reviewer #1: Yes

6. Review Comments to the Author

Reviewer #1: In general, in this new version, the manuscript has improved in quality and the presentation of findings is better. However, the authors need to mitigate their interpretation of results (especially in the conclusion of the abstract) and formally referred to the fact that they have conducted a pilot study to assess its feasibility and to explore some variables measured among children, mothers and fathers to assess the effect of a family-centered music therapy intervention. The objective of current study was NOT to provide a final conclusion of an effect of the intervention as it is stated in this new version of the manuscript. The findings would need to be further confirmed in a formal and sufficiently powered randomised controlled trial. Finally, even if this version of the manuscript has improved, some important information on the description of the new statistical methods performed is lacking. Moreover, I would suggest that this new version should be evaluated by a biostatistician.

More specific remarks follow:

1) The justification for sample size is data-driven and it seems that it has been corrected to match with the numbers used.

2) There are many exclusions in the study that are not clearly described to my opinion. What were the reasons for the family to request destruction of their data after parental withdrawal? What do you mean by “For reasons of feasibility, this design was adapted to a pre-post design during the course of the study” and did you exclude 9 data sets because they were collected after the pre-post follow-up design? Please explain.

3) In this version, you have analysed the data using a principal component analysis (PCA) in the aim to evaluate the joined effect of several variables in a limited number of dimensions; an exploratory PCA was then performed to explore the existence of a latent variable explaining the underlying variables grouped in the dimensions. The description of the statistical methods should be revised by a biostatistician as I am not a specialist of the methods used here. The description of the statistical methods used for the PCA and exploratory PCA seems lacunar (you are supposed to describe how have you chosen the number of factors retained by a scree test, then you need to apply a rotation on the factors to interpret the patterns, etc…). Some methods are presented in the results section but would be more appropriately described in the statistical methods section of the manuscript.

4) Multiple comparisons (regarding the 6 individual primary outcomes) were performed between the two randomization groups but there is no correction for the inflation of alpha error. This should be modified in a new version of the manuscript.

5) English still needs to be revised in the new version of the manuscript (e.g. line 273 of the tracked changes version: “conducted” and not “conduced”).

6) For decimals, “.” should be used and not “,”. Please correct throughout the manuscript.

7) In randomised controlled trial, comparisons of baseline characteristics between the two randomisation groups should not use statistical tests but should only be descriptive. Any differences in baseline characteristics are the result of chance rather than bias (cf. CONSORT recommendations). Tests of baseline differences are not necessarily wrong, just illogical.

8) How did you explain this change in the selection of participants in the two groups? (Table 2). As already written, multiple test comparisons lead to false positives and this needs to be corrected using appropriate statistical methods. Why did you use one-sided t tests? Two-sided tests are generally preferred in the context of superiority trials.

9) More generally, the aim of current study was to assess its feasibility and to define which variables could be used to test for an effect of the family-centered intervention. The results provided here are interesting and they motivate the conduct of a new formal and sufficiently powered randomised controlled trial. However, throughout the manuscript, there is always some confusion on the objectives of this pilot study; interpretation of study findings are always overstated and do not refer to the investigators’ primary aim: to assess if such a study is feasible and if such an intervention could provide some signal on an effect of family-centered music therapy that needs to be formally tested in a bigger trial. This is misleading and the presentation and discussion on the results need to be mitigated in order to fit with the original objective of this pilot study. In the limitations, the reduction of study power due to exclusions should not be listed as again this was a pilot study where power is by definition not sufficient.

In summary, this new version of the manuscript still requires some revisions and I would recommend that statistical methods included in this new version should be assessed by a biostatistician.

7. PLOS authors have the option to publish the peer review history of their article (what does this mean?). If published, this will include your full peer review and any attached files.

Reviewer #1: No

---

## [Author Response · Author response to Decision Letter 1]

15 Mar 2021

Reply to reviewers’ comments on the revision of

“Family-centered music therapy - empowering premature infants and their primary caregivers through music: Results of a pilot study”

We would like to thank the reviewers for their careful reading and appreciation of our revised manuscript, and their helpful comments and critiques. Please find our detailed point-by-point reply to the comments and a list of all changes made to meet the reviewers’ requests below. 

1) Sample size and participant exclusions

 “The justification for sample size is data-driven and it seems that it has been corrected to match with the numbers used.

[…]

There are many exclusions in the study that are not clearly described to my opinion. What were the reasons for the family to request destruction of their data after parental withdrawal? What do you mean by “For reasons of feasibility, this design was adapted to a pre-post design during the course of the study” and did you exclude 9 data sets because they were collected after the pre-post follow-up design? Please explain.”

We have now included a power analysis and compared the estimated sample size with the one aimed for in the current study (line 142-144). As now made more explicit in the text (line 144-146), the sample size for the power we aimed for was not feasible within the planned duration of the recruitment process, given the estimated birth rate at the hospital where the study was conducted. We emphasize that no intentional correction of the sample size was performed. The fact that exactly 50 parent-infant pairs remained for the final analysis was a coincidence. We also made efforts to document the process for participant exclusion more clearly (line 161-178).

2) Description of the statistical methods

“The description of the statistical methods should be revised by a biostatistician as I am not a specialist of the methods used here. The description of the statistical methods used for the PCA and exploratory PCA seems lacunar (you are supposed to describe how have you chosen the number of factors retained by a scree test, then you need to apply a rotation on the factors to interpret the patterns, etc…). Some methods are presented in the results section but would be more appropriately described in the statistical methods section of the manuscript.”

We agree that the full details on the PCA should be provided in the methods section. We have added such a description (line 274-279) and removed these details from the results section. Furthermore, the description of all statistical methods has been checked and approved by two statisticians who were not involved in the study. Following their advice, we have removed the first PCA extracting a single factor, as it adds no further insight compared to the multivariate comparison and may have led to confusion. 

3) Correction for multiple comparisons

“Multiple comparisons (regarding the 6 individual primary outcomes) were performed between the two randomization groups but there is no correction for the inflation of alpha error. This should be modified in a new version of the manuscript.”

Indeed, such a correction is necessary. We apologize for this objective error. We applied a Bonferroni-Holm correction of the p values on all post-hoc comparisons (line 272-273 and Table 2). Actually, this correction renders all these comparisons insignificant, while the comparison for the two factors still remains valid. We have mitigated our conclusions accordingly (see below).

4) Language editing

“English still needs to be revised in the new version of the manuscript (e.g. line 273 of the tracked changes version: “conducted” and not “conduced”).

[…]

For decimals, “.” should be used and not “,”. Please correct throughout the manuscript.”

We have asked a person that was not involved in the study to carefully proofread the manuscript and hope that there are no errors left to disturb the flow of reading. The decimals have been corrected.

5) Baseline characteristics

“In randomised controlled trial, comparisons of baseline characteristics between the two randomisation groups should not use statistical tests but should only be descriptive. Any differences in baseline characteristics are the result of chance rather than bias (cf. CONSORT recommendations). Tests of baseline differences are not necessarily wrong, just illogical.

[…]

How did you explain this change in the selection of participants in the two groups? (Table 2).”

We have removed these statistical tests from Table 1. Regarding the different participant numbers in Table 1 and 2, note that we have initially presented the baseline characteristics of the whole sample (65 parent-infant pairs, minus one pair who withdraw from the study, c.f. line 165-166), while in Table 2, only the data of the final sample (50 pairs) was reported. To avoid confusion, we now report baseline characteristics of the final sample in Table 1.

6) One-sided t test

“Why did you use one-sided t tests? Two-sided tests are generally preferred in the context of superiority trials.”

We have used one-sided tests to be consistent with our hypotheses, which were directed: An increase in the developmental parameters and a decrease in therapy duration and parental stress is expected in the intervention group, not a mere difference from the control group. This justification has been made explicit in the methods section (lines 280-283). We are aware of the convention to use two-sided tests in superiority trials (which the study does not even explicitly claim to perform). However, this convention is debated in the statistical literature and, in our opinion, is simply illogical, as a two-sided test would render a strongly negative effect of music therapy significant as well, which is at odds with its purpose to test an explicitly positive effect.

7) Objectives and conclusions of the study

“throughout the manuscript, there is always some confusion on the objectives of this pilot study; interpretation of study findings are always overstated and do not refer to the investigators’ primary aim: to assess if such a study is feasible and if such an intervention could provide some signal on an effect of family-centered music therapy that needs to be formally tested in a bigger trial. This is misleading and the presentation and discussion on the results need to be mitigated in order to fit with the original objective of this pilot study. In the limitations, the reduction of study power due to exclusions should not be listed as again this was a pilot study where power is by definition not sufficient.”

The manuscript was completely revised: Attention was paid to ensure that the presentation of the objectives of the study and the presentation of the results and their interpretation now fit together more closely. The interpretation of the results was significantly mitigated in order to fit with the objectives of the study and to avoid misleading interpretation or overinterpretation.

---

## [Editor Report · Decision Letter 2]

31 Mar 2021

Family-centered music therapy - empowering premature infants and their primary caregivers through music: Results of a pilot study.

PONE-D-20-11446R2

Dear Dr. Menke,

We’re pleased to inform you that your manuscript has been judged scientifically suitable for publication and will be formally accepted for publication once it meets all outstanding technical requirements.

Kind regards,

Olivier Baud, MD, PhD

Academic Editor

PLOS ONE
---

## [Editor Report · Acceptance letter]

6 May 2021

PONE-D-20-11446R2 

Family-centered music therapy - empowering premature infants and their primary caregivers through music: Results of a pilot study 

Dear Dr. Menke:

I'm pleased to inform you that your manuscript has been deemed suitable for publication in PLOS ONE. Congratulations! Your manuscript is now with our production department. 

Kind regards, 

on behalf of

Pr. Olivier Baud 

Academic Editor

PLOS ONE